# Biodegradable polyphosphoester micelles act as both background-free $^{31}$P magnetic resonance imaging agents and drug nanocarriers

Olga Koshkina [1,6] ✉, Timo Rheinberger [1,6], Vera Flocke[2], Anton Windfelder [3,4], Pascal Bouvain [2], Naomi M. Hamelmann [5], Jos M. J. Paulusse [5], Hubert Gojzewski[1], Ulrich Flögel [2] ✉ & Frederik R. Wurm [1] ✉

In vivo monitoring of polymers is crucial for drug delivery and tissue regeneration. Magnetic resonance imaging (MRI) is a whole-body imaging technique, and heteronuclear MRI allows quantitative imaging. However, MRI agents can result in environmental pollution and organ accumulation. To address this, we introduce biocompatible and biodegradable polyphosphoesters, as MRI-traceable polymers using the $^{31}$P centers in the polymer backbone. We overcome challenges in $^{31}$P MRI, including background interference and low sensitivity, by modifying the molecular environment of $^{31}$P, assembling polymers into colloids, and tailoring the polymers' microstructure to adjust MRI-relaxation times. Specifically, gradient-type polyphosphonate-copolymers demonstrate improved MRI-relaxation times compared to homo- and block copolymers, making them suitable for imaging. We validate background-free imaging and biodegradation in vivo using *Manduca sexta*. Furthermore, encapsulating the potent drug PROTAC allows using these amphiphilic copolymers to simultaneously deliver drugs, enabling theranostics. This first report paves the way for polyphosphoesters as background-free MRI-traceable polymers for theranostic applications.

Monitoring polymers in vivo plays a pivotal role in the development of materials for biomedical applications, such as drug delivery and tissue regeneration[1–4]. MRI is a powerful $^1$H-based anatomical imaging technique, which is free of ionizing radiation and not limited by tissue penetration depth. Heteronuclear "hot spot" MRI opened new

horizons, introducing the imaging of other nuclei, currently focusing on $^{19}$F[5–7]. While conventional $^1$H MRI agents modulate tissue contrast, heteronuclear agents are highly specific and act as new color, uniquely providing quantitative information along with the spatial localization and monitoring over time. Therefore, "hot spot" $^{19}$F MRI became

[1]Sustainable Polymer Chemistry Group, Department of Molecules and Materials, MESA+ Institute of Nanotechnology, Faculty of Science and Technology, University of Twente, Enschede, The Netherlands. [2]Department of Molecular Cardiology, Experimental Cardiovascular Imaging, Heinrich Heine University, Düsseldorf, Germany. [3]Department of Bioresources, Fraunhofer Institute for Molecular Biology and Applied Ecology IME, Giessen, Germany. [4]Laboratory of Experimental Radiology, Justus Liebig University, Giessen, Germany. [5]Biomolecular Nanotechnology Group, Department of Molecules and Materials, MESA+ Institute of Nanotechnology, University of Twente, Enschede, The Netherlands. [6]These authors contributed equally: Olga Koshkina, Timo Rheinberger. ✉e-mail: o.koshkina@utwente.nl; floegel@uni-duesseldorf.de; frederik.wurm@utwente.nl

effective in imaging of cardiovascular diseases, cancer, monitoring cellular therapies, nanomedicines, and biomaterials, such as artificial tissues[8–14]. However, common [19]F MR agents are formulations of per- and polyfluoroalkyl substances (PFAS or perfluorocarbon PFC), which are under discussion recently[15]. Here, we introduce an alternative solution based on biocompatible and biodegradable polymers, i.e., polyphosphoesters, opening the route to more sustainable MRI-traceable polymer materials.

Currently, polymeric materials, such as nanocarriers and hydrogels can be labeled for heteronuclear MRI by encapsulation of PFAS. However, PFAS are hydrophobic, lipophobic (amphiphobic), and chemically inert, which makes their stabilization in physiological media and functionalization challenging[6,16]. Moreover, PFAS often show prolonged organ deposition, which is a disadvantage for example in repeated imaging sessions and can lead to misdetection, e.g., when the labeled material itself was degraded[17]. Lastly, due to the high chemical stability which makes PFAS basically undegradable, they are classified as environmental pollutants[15]. Current approaches focus on improving the properties of fluorinated compounds for [19]F MRI, e. g. developing of partly fluorinated polymers which are stable in physiological milieu, or inorganic nanoparticles[13,18–25]. Instead, we decided to develop polymers that can be imaged via [31]P nuclei in their backbone instead of common [19]F, to take advantage of the biocompatible properties of polyphosphoesters[26,27].

[31]P is the natural, 100% abundant, NMR-active isotope of phosphorus. However, the development of background-free [31]P MRI agents and [31]P MRI-traceable materials has been hampered by several factors, including the intrinsic background from natural phosphates, the low MR sensitivity of [31]P of 7% compared to [1]H, and other unfavorable MR characteristics, including too short or too long relaxation times and J coupling. For example, most of biomolecules yield weak [31]P NMR signals due to these reasons. Nevertheless, [31]P MR spectroscopic approaches were used for pH-sensing and detection of metabolites, such as endogenous phosphocreatine and ATP, as well as non-biodegradable synthetic polymers[28–30], but are limited due to the required long acquisition times.

Here, we introduce biodegradable nanocarriers that are made of bioinspired polyphosphoesters with a high [31]P content and variable chemical shifts to overcome the low sensitivity of [31]P MRI. PPEs are versatile polymers with highly diverse chemistry, which make them interesting candidates for several biomedical applications, particularly as a stealth coating[26,27]. Uniquely, we synthesized biodegradable, [31]P MRI agents based on amphiphilic gradient copolymer micelles which can encapsulate hydrophobic therapeutic payload as their second inherent function, acting as both background-free imaging agents and carrier material for drugs. This work will lead to the development of MRI-traceable biomedical polymers for a broad range of applications, from MR imaging agents to label-free [31]P MRI-traceable drug delivery and materials for tissue engineering.

## Results and discussion
### Proof of concept that polyphosphonate nanocarriers can be imaged with [31]P MRI
We selected the following strategy to solve the fundamental challenges in MRI of [31]P:

(1) to ensure the background-free imaging, we tuned the molecular environment of [31]P. Specifically, we used polyphosphonates (PPn, general structure Fig. 1A), which contain a P-C-bond, which is not present in endogenous phosphates resulting in a chemical shift that is separated by more than 10 ppm from other biomolecules to allow selective and specific imaging. The shift can be further adjusted by varying the chemical nature of the side chain in Fig. 1A, generating different resonance frequencies and thereby give access to additional MRI 'colors'.

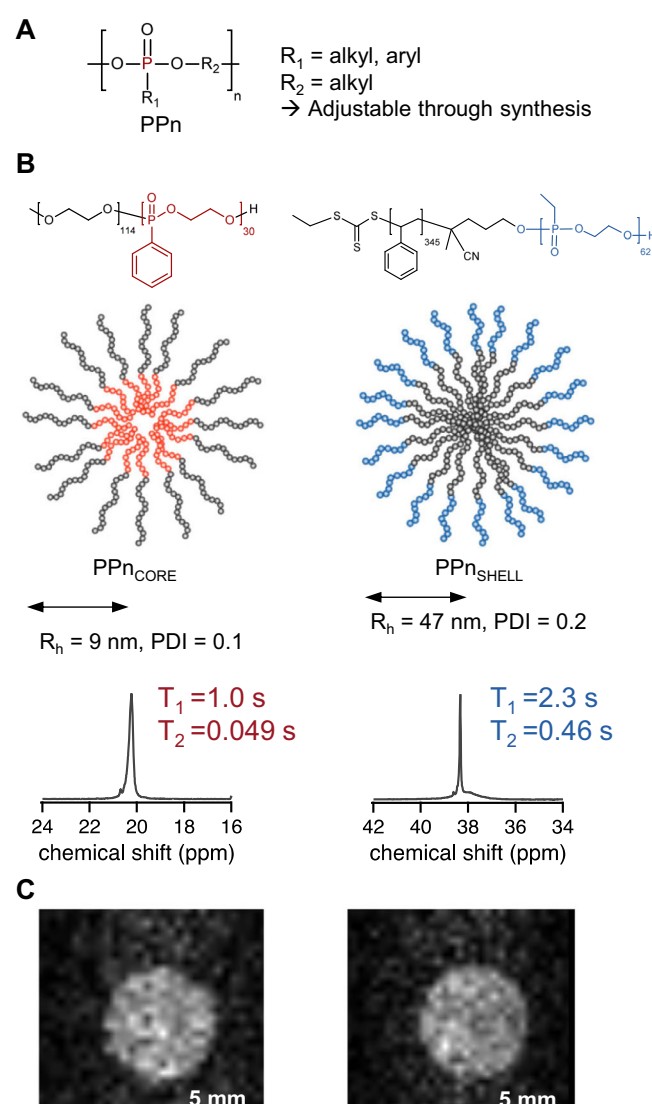

**Fig. 1 | Proof-of-concept that PPn colloids can be used in [31]P MRI.** (A) General chemical structure of PPn. The composition of the main and side chains can be adjusted to tune the properties. (B) Chemical structures of block copolymers and schematic of colloidal nanostructures PPn_CORE that contains phosphorus-31 in the core (left) PPn_SHELL (right) with phosphorus-31 in hydrophilic PPn shell, and the respective NMR spectra. The relaxation times were suitable for imaging (NMR spectroscopy in $H_2O/D_2O$ 9:1 $v:v$, decoupled). (C) [31]P MR images of aqueous dispersions of PPn_SHELL and PPn_CORE, (polymer concentration 4 wt.-% polymer correspond to 0.1 M [31]P for PPn_CORE and 0.05 M [31]P in PPn_SHELL), TAcq 17 min, FOV 20×20 mm², matrix 64×64, 9.4 T.

(2) We prepared PPE-copolymers that can self-assemble into colloidal nanostructures to enhance the local [31]P content of within the agent, as the MR signal linearly correlates with the local concentration of [31]P.

(3) We developed polymers with a low glass transition temperature ($T_g$ < body temperature) to adjust the MR relaxation times in a favorable manner: Usually, a shorter transverse relaxation time $T_1$ and a longer longitudinal relaxation time $T_2$ lead to higher signal-to-noise ratios (SNR)[31]. However, solid polymers typically display a very short $T_2$, but polymers above the $T_g$ are in a "liquid-like" and not in a solid state, which should enhance both $T_2$ and SNR.

Our first goal was to obtain proof-of-concept that [31]P MRI is possible with colloidal PPn nanostructures despite the low sensitivity of

the $^{31}$P nucleus. Our first attempt were nanoparticles prepared by an emulsion formulation utilizing hydrophobic polyphosphonates, which, however, displayed a too short $T_2$ of 0.017 s (Supplementary Fig. 1). To adjust the relaxation times, we expanded to amphiphilic block-copolymer microstructures. We focused on polymers with a $^{31}$P-free MR-invisible second block (Fig. 1B) to obtain a single resonance signal for conventional spin-echo MRI sequences. For this strategy, we synthesized an amphiphilic block copolymer of hydrophobic PPn with phenyl side groups (Ph-PPn) and hydrophilic polyethylene glycol that self-assemble into micelles in aqueous solution with PPn in the core (PPn$_{CORE}$, Fig. 1B, Supplementary Table 1). Additionally, we prepared poly(styrene-b-ethyl phosphonate)[32] with the hydrophilic PPn with ethyl side chains (Et-PPn) in the shell of the micelles PPn$_{SHELL}$ (Fig. 1B, Supplementary Table 1). The hydrophobic PPn$_{CORE}$ already displayed a longer $T_2$ of 0.049 s in micelles. PPn$_{SHELL}$ showed an even higher $T_2$ due to the higher mobility of the hydrophilic Ethyl-PPn. As a result of improved $T_2$, both nanostructures were successfully imaged with $^{31}$P MRI using a spin-echo sequence within acquisition time of 17 min (TAcq; Fig. 1C). These results provided the first proof that polyphosphonates can be imaged with MRI.

### Gradient microstructures improve MRI characteristics

Based on these initial results, we hypothesized that a gradient microstructure could further beneficially affect the MR properties, as it should enable an even higher mobility of the macromolecules resulting in longer $T_2$ compared to block copolymer with a single phase boundary. Moreover, the $^{31}$P content and consequently the sensitivity should increase by copolymerization of two $^{31}$P-containing monomers. Therefore, we developed a single-step synthesis approach to amphiphilic $^{31}$P MRI agents by anionic ring-opening copolymerization of a mixture of phenyl- and ethyl-functional monomers (Fig. 2A, Supplementary Table 1).

The resulting copolymers with Ph-PPn and Et-PPn repeat units displayed a gradient microstructure, as confirmed by both kinetic measurements (Fig. 2B, Supplementary section 1.3,) and Monte Carlo simulations (Fig. 2C, D, further characterization Supplementary Figs. 3, 4). In aqueous dispersion, they self-assembled to micelles PPn$_{GRAD}$ of 9 nm radius with a narrow size distribution (PDI < 0.1) and a low critical micelle concentration of 14 mg L$^{-1}$ similar to conventional nonionic surfactants (Fig. 2E, see Supplementary Figs. 5−9 for further characterization including stability and atomic force microscopy images).

Indeed, the gradient structure led to improved MR properties (Fig. 2E, F, cf. Supplementary Table 3 for relaxation times of different batches). The $T_2$ of hydrophilic Et-PPn-block was twice as long compared with the hydrophilic shell in the above-mentioned block copolymer micelles PPn$_{SHELL}$ (0.9 s versus 0.5 s; see Supplementary Section 2.1 for fitting and discussion). Similarly, the $T_2$ of Ph-PPn in the core increased to 0.13 s. As the gradient microstructure creates a broader interfacial region in the micelles, the mobility of macromolecules and the $T_2$ of hydrophobic core increase compared to the other nanostructures. Furthermore, both $^{31}$P-containing blocks lead to a higher $^{31}$P content in a micelle of 20 wt-%, which along with longer $T_2$ should lead to higher SNR when a similar polymer dose is injected.

To take full advantage of both the increased $T_2$ and $^{31}$P concentration in the micelles, we implemented and optimized multi-chemical selective imaging (mCSSI; spin-echo type of sequence) for $^{31}$P MRI. Usually, conventional MRI of compounds with several NMR signals results in chemical shift artefacts, making a reliable signal assignment and localization impossible. The mCSSI sequence allows for simultaneous, artefact-free imaging of multiple signals and subsequent summation of the individual images that results in increased SNR[9]. Indeed, mCSSI of the different PPn-colloids at the same polymer dose in aqueous dispersion revealed a three-fold

higher SNR for the PPn$_{GRAD}$ micelles compared to the both block-copolymers (Fig. 2F, TAcq 17 min). The signal intensity decreased linearly, when the PPn$_{GRAD}$ dispersion was diluted (Supplementary Fig. 11). A comparison to $^{19}$F MRI using an emulsion of perfluoro-15-crown 5 ether using 3.8 mol L$^{-1}$ $^{19}$F, vs 1.5 mol L$^{-1}$ $^{31}$P showed approx. 5 times higher SNR for $^{19}$F (Supplementary Fig. 11). However, these samples are not directly comparable due to their different physicochemical properties. The lowest detectable concentration of polymer was comparable to the one reported for fluorinated polymers previously[13], indicating sufficient signal intensity for in vivo studies by $^{31}$P MRI.

To demonstrate that gradient micelles can be reliably imaged in tissues, we injected PPn$_{GRAD}$ micelles via the stem base into a physalis berry (Fig. 2G left, arrow). $^{31}$P mCSSI led to artefact-free images without endogenous background and merge with the 'anatomical' $^1$H MRI demonstrated the distribution of PPn$_{GRAD}$ micelles from their injection site into the core of the physalis (Fig. 2G).

### In vivo $^{31}$P MRI and biodegradation of polyphosphonate micelles

After encouraging $^{31}$P MRI in vitro, we confirmed that $^{31}$P mCSSI of gradient micelles PPn$_{GRAD}$ can be used in vivo. We used *Manduca sexta* caterpillars (Fig. 3A); *M. sexta* has a hemolymph volume of 1−2 mL, comparable to the blood volume of mice, which makes it a suitable, alternative animal imaging model instead of mammalians according to 3 R principles of animal-friendly testing[33,34].

First, we demonstrated that PPn$_{GRAD}$ micelles can be imaged during the circulation in hemolymph (i.e., the blood of insects). Therefore, the micelles were injected into the dorsal vessel of the animals. In spite of the lower sensitivity of the $^{31}$P nucleus compared to $^{19}$F, we selected an overall colloid dose typical for intravenous injection of $^{19}$F MRI agents (15 mg in 100 μL). All animals tolerated these injections well and continued their development as usual. The micelles distributed in the hemolymph homogenously after injection, as shown by overlaying the morphological $^1$H MRI and $^{31}$P mCSSI data (Fig. 3C, Supplementary Fig. 12, TAcq 17 min). After 24 h, the $^{31}$P signal was still located in the hemolymph and decreased slightly (Fig. 3C). Thus, only small amounts of the polymers were excreted or degraded under these conditions; the biodistribution should be studied in the future in different animal models. We assume that the previously reported stealth effect of Et-PPn prolonged the circulation time, allowing one to detect the micelles in hemolymph after 24 h similar to PEGylated nanocarriers[35].

Finally, we confirmed the biodegradation of the PPn$_{GRAD}$ micelles in the *M. sexta* model. The agents were directly injected into the anterior part of the gut; $^{31}$P and $^1$H MRI confirmed the localization of the agents within the target organ (Fig. 3E). After 24 h, we collected the feces of these animals and used $^{31}$P NMR spectroscopy to analyze the degradation products (Fig. 3F). Indeed, the $^{31}$P NMR spectrum of the aqueous extracts proved the presence of characteristic sharp resonances of low molar mass degradation products at 16 and 31 ppm next to the broader signals of the injected agent at 20 and 38 ppm, respectively, with the chemical shifts as expected for the two degradation products (Fig. 3F, see Supplementary Fig. 13 for further animals). In *M. sexta*, the gut passage takes ca. 1.5 h, which is obviously insufficient for complete biodegradation. PPEs degrade mainly by a backbiting mechanism[36]; as Et-PPn is the outer block, it degrades first, followed by Ph-PPn, thereby resulting in a stronger signal of degradation product of Et-PPn (Fig. 3F). Hence, these results confirm that PPn$_{GRAD}$ agents are degradable in vivo.

### Polyphosphonate micelles act as drug nanocarriers

Besides in vivo $^{31}$P MRI, we proved that PPn$_{GRAD}$ micelles can be used to formulate hydrophobic therapeutics for drug delivery, which is currently the main biomedical application of amphiphilic copolymer micelles[37]. We selected Proteolysis Targeting Chimera (PROTAC, ARV-

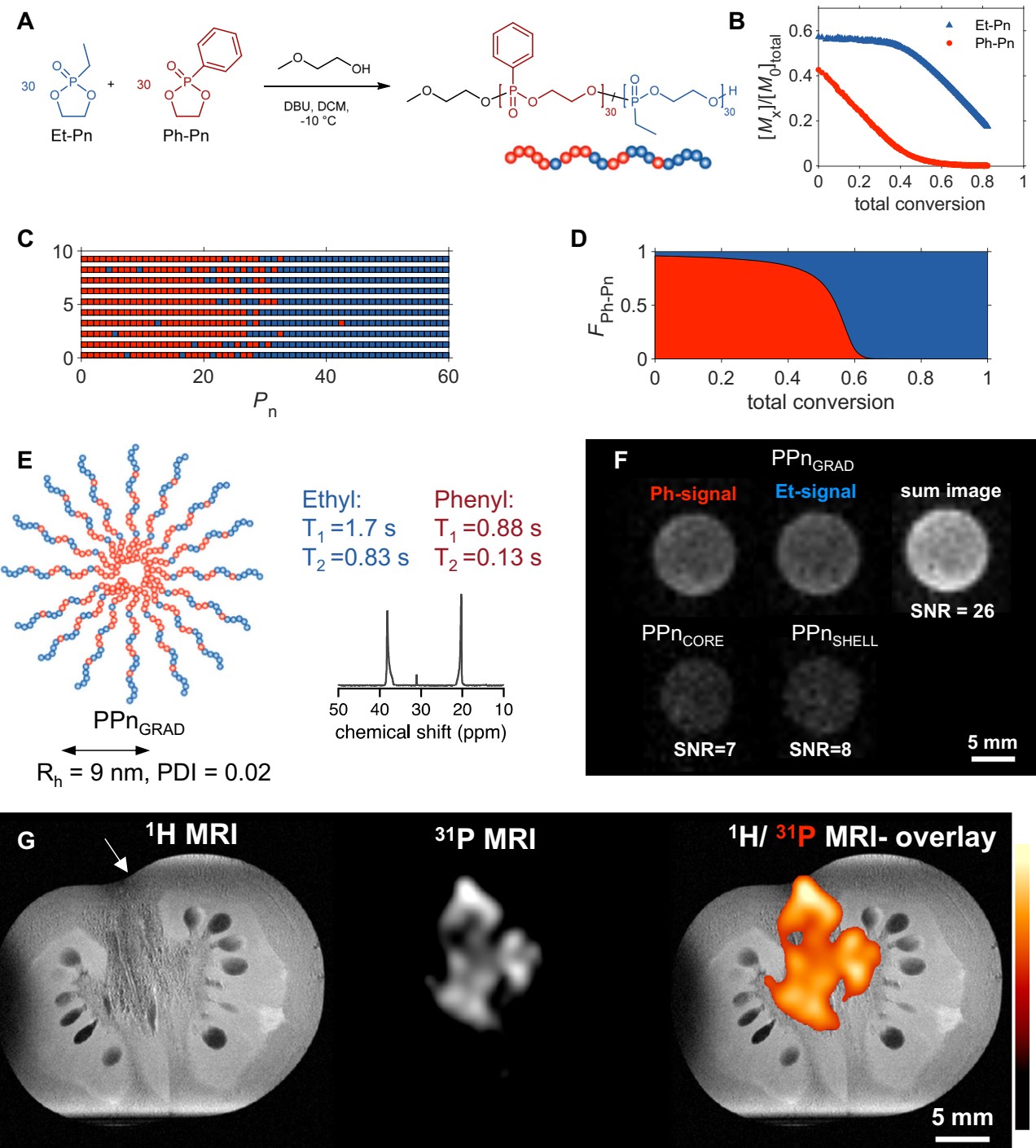

**Fig. 2 | Synthesis and $^{31}$P MRI of gradient copolymer micelles PPn$_{GRAD}$.**
**A** Synthesis of gradient copolymer by anionic ring-opening copolymerization.
**B** Copolymerization kinetics: Monomer concentration in solution as a function of
the total conversion of monomers Et-Pn and Ph-Pn. **C** Monte-Carlo simulated
microstructure for 10 representative polymer chains, calculated from the deter-
mined reactivity ratios. **D** Average composition of monomer fraction $F$ plotted
against total conversion. **E** Schematic of PPn$_{GRAD}$ micelles with NMR spectrum
(158 MHz, D$_2$O). **F** $^{31}$P MRI of PPn$_{GRAD}$ micelles (top row). The signals of both Ph- and
Et-units were added to a sum image, resulting in the highest SNR. The images of

block copolymer micelles PPn$_{CORE}$ and PPn$_{SHELL}$ are shown at the same intensity
scale for comparison. Polymer content 4 wt% ($^{31}$P 0.1 M in PPn$_{CORE}$, 2x0.1 M in each
block of PPn$_{GRAD}$; and 0.05 M $^{31}$P In PPn$_{SHELL}$; similar polymer concentration was
chosen. FOV 20 × 20 mm$^2$, matrix 64 × 64, 9.4 T. **G** PPn$_{GRAD}$ micelles could be
imaged and localized with $^{31}$P MRI after injection in physalis berry. $^1$H MRI (matrix
512 × 512), $^{31}$P MRI and an overlay image ($^{31}$P MRI in hot iron, matrix 64 × 64) are
shown. Arrow indicates the injection site, mCSSI, TAcq 17 min, FOV 30 × 30 mm$^{2,}$
9.4 T. Color scale in arbitrary units (a.u.).

825) as a model payload[38]. PROTACs are emerging drugs that induce
the degradation of pathogenic proteins by the cellular degradation
machinery, holding the potential to treat currently undruggable tar-
gets, such as resistant tumors[38]. However, many PROTACS are large

and poorly water-soluble molecules that need to be formulated to be
stable in body fluids.

We used a simple nanoprecipitation technique to formulate
PROTAC-loaded PPn$_{GRAD}$ micelles. The resulting aqueous dispersion

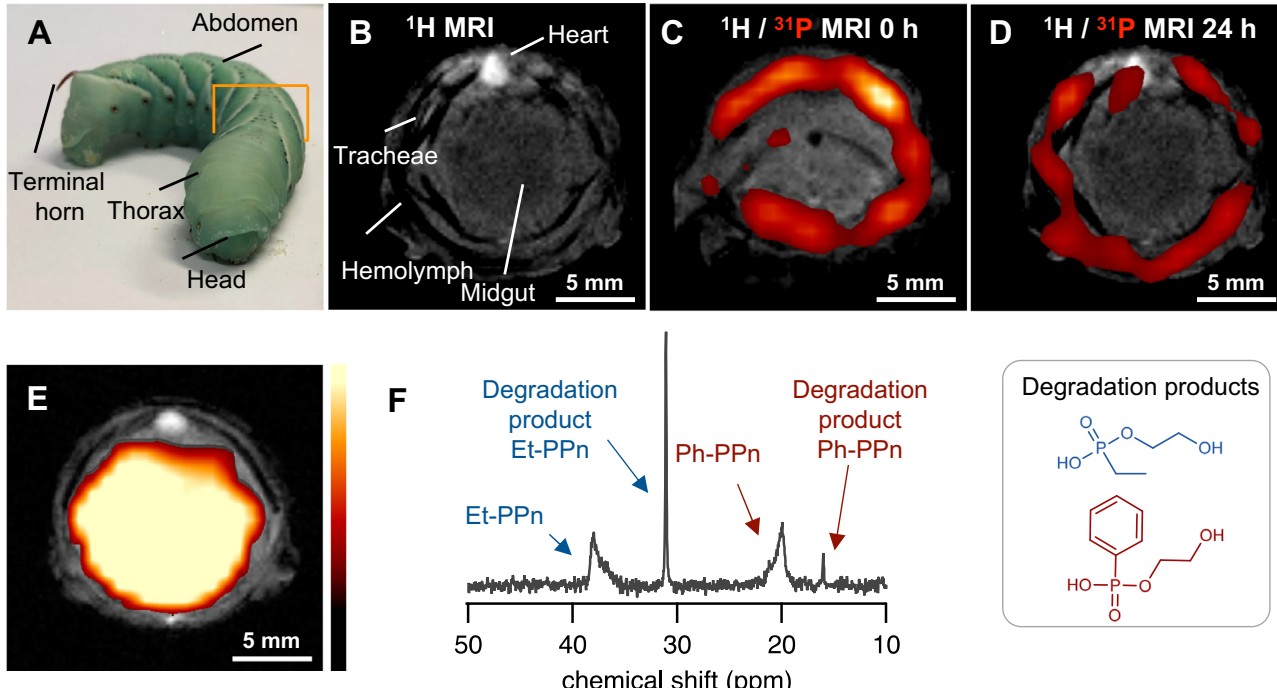

**Fig. 3 | ¹H/³¹P MRI in *Manduca sexta* caterpillars in vivo and degradation of the agents (FOV 20 × 20 mm², ¹H MRI matrix 256 × 256, ³¹P MRI matrix 64 × 64, 9.4 T). A** Photograph of the *M. sexta* caterpillar, orange square denotes the region shown on MR images. **B** ¹H MRI showing the anatomy of caterpillar. **C** ³¹P MRI overlaid on ¹H MR image after the injection of the agents into the dorsal vessel (heart). PPn$_{GRAD}$ agents circulate in hemolymph. **D** After 24 h the agents can still be found in the circulation, as expected for micelles with stealth surface. **E**, **F** Degradation of PPn$_{GRAD}$ in vivo. **E** After injection of the agents into the gut, overlay of ³¹P MR on ¹H MR image confirms their localization. **F** ³¹P NMR spectrum of feces (D$_2$O, 158 MHz) collected after 24 h shows the characteristic signals of degradation products confirming that PPn$_{GRAD}$ agents are biodegradable. Color scale in a.u.

appeared translucent, with micelles of 10 nm radii similar to the non-loaded micelles, as determined by DLS. The aqueous dispersion was stable over at least two months according to DLS (Fig. 4A right vial and 4B). Conversely, when PROTAC was formulated without the amphiphilic polymer, the hydrophobic drug precipitated and formed polydisperse aggregates immediately after the addition of the organic solution into water (Fig. 4A left vial, and 4B). The PROTAC-PPn$_{GRAD}$ micelles reduced the viability of cancer cells (HeLa Fig. 4C) and induced early apoptosis (Fig. 4D) similar to free PROTAC, suggesting that PPn$_{GRAD}$ can be to used formulate hydrophobic drugs. Overall, these results suggest that PPn$_{GRAD}$ can be used to develop theranostic agents in the future.

To conclude, we developed biocompatible, biodegradable micelles for ³¹P MRI based on polyphosphoesters, overcoming the low sensitivity of the ³¹P MRI. Therefore, we have synthesized and compared different colloidal nanostructures. Based on these results we have developed amphiphilic gradient copolymers using polyphosphonates, which self-assemble into well-defined micelles with advantageous ³¹P MRI characteristics. The PPE micelles were injected into *M. sexta* and successfully imaged in vivo using ³¹P MRI. The in vivo biodegradation of the PPE-micelles was underlined from collected feces by ³¹P NMR. These amphiphilic micelles can encapsulate and deliver hydrophobic payload as their second inherent function in parallel to their function as ³¹P MRI agents, as we have exemplarily shown for PROTAC. This unique polyphosphoester platform acts simultaneously as the carrier material for therapeutics and as the imaging agent and we believe pave the way for future MRI-traceable polymers.

## Methods

**Size exclusion chromatography (SEC)** measurements were performed in DMF (containing 1 g·L⁻¹ of LiBr) at 60 °C and a flow rate of 1 mL min⁻¹ with a PSS SECcurity as an integrated instrument, including three PSS GRAM column (100/1000/1000 g mol⁻¹) and a refractive index (RI) detector. Calibration was carried out using polystyrene standards supplied by Polymer Standards Service. The SEC data were plotted with OriginPro 9 software from OriginLab Corporation.

**NMR spectroscopy** was measured at Brucker Avance III 400 MHz spectrometer equipped with a PA BBO 400S1 BBF-H-D-05 Z SP probe at 298 K (driven by TopSpin 8). As deuterated solvents CDCl$_3$, CD$_2$Cl$_2$, or D$_2$O were used. The proton spectra were calibrated against the solvent signal (CDCl$_3$: δ H = 7.26 ppm, CD$_2$Cl$_2$: δ H = 5.32 ppm, D$_2$O: δ H = 4.79 ppm). Longitudinal relaxation times T$_1$ were measured using an inverse recovery sequence. For the measurements of transverse relaxation times T$_2$ the Carr-Purcell-Meiboom-Gill (CPMG) sequence was used. At least 10 data points were acquired, which were then used for data fitting. An interscan delay d1 was set to 5 × T$_1$. In case of samples in water, deuterium oxide (10 vol%) was added to the samples for locking. Data analysis was performed using Mestrenova14 from Mestrelab and OriginPro2019b.

**Dynamic light scattering** (DLS) was done at a Zetasizer Lab from Malvern UK at a scattering angle of 90°, and 295 K. The samples were diluted with ultrapure water or physiological saline so that the attenuator was at step 10-11 (set automatically by the device). Data analysis was done with ZSxplorer 2.2.0.147 software from Malvern Panalytical.

**Fluorescence spectroscopy** to determine the critical micelle concentration (CMC) using *pyrene assay*[39] was measured at spectrometer FL6500 spectrometer from Perkin Elmer, equipped with a Pulse Xenon lamp. A concentration series of polymer in ultrapure water between c = 1.1 mg mL⁻¹ and 7.7 × 10⁻⁵ mg mL⁻¹ was prepared and mixed with pyrene stock solution in methanol (1.2 × 10⁻⁴ mol L⁻¹) so that the final concentration of pyrene was 6 × 10⁻⁷ mol L⁻¹;

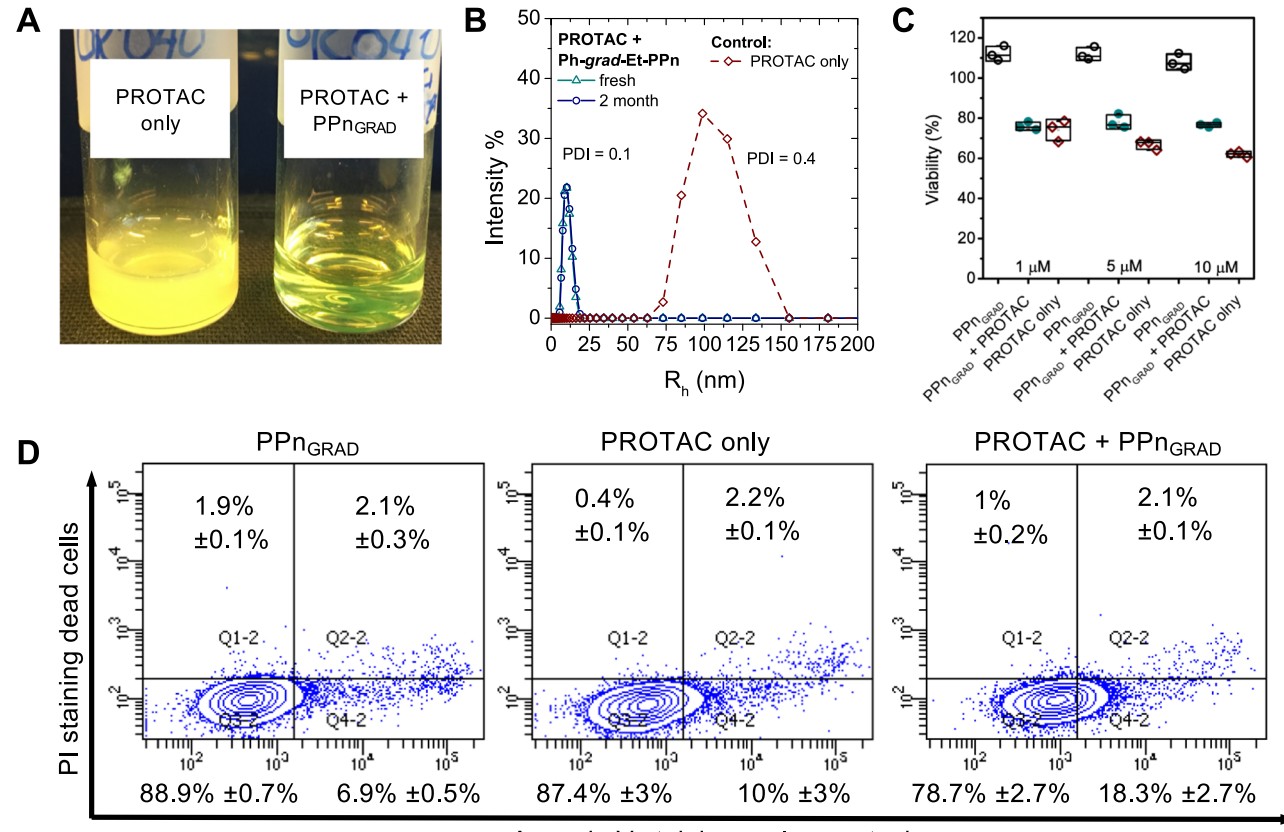

**Fig. 4 | Stabilization of hydrophobic payload by PPn_GRAD ³¹P MRI agents applied to PROTAC ARV-825. A** Photograph of PROTAC-loaded PPn_GRAD micelles (right vial) and a control PROTAC suspension prepared without the addition of PPn_GRAD (left vial). PROTAC-loaded micelles form a clear dispersion. Conversely, in the control PROTAC (yellow) precipitated immediately after the preparation. **B** DLS of PROTAC-loaded PPn_GRAD micelles. The micelles had a low polydispersity and were stable for at least two months. In contrast, PROTAC suspension displayed a broad size distribution. The size measurement of PROTAC-only suspension can be affected by sedimentation. **C** Viability of cancer cells (HeLa) was reduced upon the administration of PROTAC-loaded micelles to a comparable level as with free PROTAC. Unloaded PPn_GRAD micelles were used as a control. The viability values are normalized to untreated controls. $n = 3$ (data presented as mean ± SD). **D** Flow cytometry shows that cancer cells treated with PROTAC-loaded micelles (1 μM ARV-825) start entering an early apoptosis phase.

typically 5 μL of pyrene stock solution were added to 995 μL of the polymer solution. The emission spectra were recorded approximately 15 min after preparation of the solutions using an excitation wavelength of 333 nm, and emission range of 350–550 nm, scan speed of 50 nm min⁻¹. To calculate the CMC, the ratio of intensities $I_1$ (372 nm) $I_3$ (383 nm) was plotted versus the logarithm of concentration $ln(c)$. The CMC was obtained as an intersection of linear fits of the low c region, where $I_1/I_3 \approx const$, and the region where $I_1/I_3$ decreases.

For **atomic force microscopy (AFM)**, the micelles were deposited from a solution (0.2 mg mL⁻¹) on freshly prepared silicon wafers by spin-coating (2000 rpm, 60 s). Silicon wafers were cleaned at two bath steps: ultrasonication in acetone and Piranha solution treatment for 20 min. AFM images were obtained in the air and at room temperature using a MultiMode 8 AFM instrument with a NanoScope V controller (Bruker). The AFM was operated in the PeakForce Quantitative Nanomechanical Mapping mode (PF-QNM) to record force-distance curves and to further processed them in the NanoScope Analysis software (version 1.9). The ScanAsyst setting was set to "on" in order to apply optimized scanning parameters, particularly the feedback loop and the applied load, for imaging soft micelles. The force-distance curves were collected following a sine-wave sample-tip trajectory with a frequency of 2 kHz and utilizing a peak-force amplitude value of 30 nm. Soft AFM cantilevers were chosen with a nominal spring constant of 0.4 N/m and a tip with a nominal radius of 2 nm (Bruker, ScanAsyst-air). The AFM optical sensitivity (deflection sensitivity) was calculated based on the thermal tune method based on the nominal spring constant[40].

**Differential scanning calorimetry (DSC)** measurements were performed using a Trios DSC 25 series thermal analysis system, running with the Trios DSC Software 5. The temperature range was from −80 °C to 50 °C under nitrogen and heating rate was 10 °C min⁻¹. All glass transition temperatures ($T_g$) were obtained from the second heating ramp of the experiment.

### Reporting summary

Further information on research design is available in the Nature Portfolio Reporting Summary linked to this article.

### Data availability

All processed data are available in the manuscript or supplementary information. Raw data and materials are available upon request.

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

## Acknowledgements

We thank Dr. Diego Resendiz Lara (UT), Dr. Ricardo E.P. Martinho (UT), Dr. Mark Hempenius (UT), Dr. Sandra S.M.C. Michel-Souzy (UT), Bianca Ruel (UT), Clemens Padberg (UT), Ramon Ten Elshof (UT), Richard Egberink (UT), and Prof. Dr. Andreas Vilcinskas (IME). We acknowledge Alexander von Humboldt Foundation (OK), Dutch Research Council NWO grant OCENW.XS21.2.066 (OK), Deutsche Forschungsgemeinschaft DFG grant INST 208/764-1 FUGG (UF).

## Author contributions

Conceptualization: O.K., F.R.W. Investigation: O.K., T.R., V.F., A.W., P.B., N.H., H.G., U.F., F.R.W.

## Funding

## Competing interests

The authors declare no competing interests.
