## [Peer Review File · Nature Communications]

Biodegradable Polyphosphoester Micelles Act as Both Background-free ^{31}P Magnetic Resonance Imaging Agents and Drug NanocarriersReviewers' Comments:

Reviewer #1:

Remarks to the Author:

The work from Koshkina et al. is about the development of biodegradable polyphosphonate micelles for ³¹P-MRI. These polymers form in aqueous solution small micelles and a tailored design of the polymer skeleton with a low glass transition T_g guarantee T₁ and T₂ suitable for imaging. This work is extremely innovative and although preliminary the smart chemistry linked to the advanced acquisition methods allow to overcome the sensitivity problems usually associated with ³¹P-MRI. This work can open up a new alternative/complementary method for in vivo imaging to ¹⁹F-MRI, which is needed as most of the perfluorinated compounds used as MRI probes can raise sustainability issues. This work deserves publication in Nature Communication after minor revisions reported below.

- it would be important to report at least for the best polymer (gradient) a ³¹P-MRI concentration study for $c > c_{mc}$ of the micellar suspension to show linearity of the signal and also the minimal concentration for which is possible to obtain a suitable SNR.
- it would be important for the reader to know the number of ³¹P atoms/mL for the reported MRI experiments rather than the w/V concentration of the polymer (also for a better comparison with ¹⁹F-MRI). They comment that they used the same "concentration" as for the probe in ¹⁹F-MRI, but it is not clear if it is the same P/F concentration or the same polymer concentration, which would be not relevant as the M_w is different (moreover, most ¹⁹F-MRI are molecular probes). This should be better explained in the text.
- Did they perform ¹⁹F-MRI measurements in Manduca for comparison? If they have them it would be important to introduce them in the ESI.
- these micelles are very small, for which application the authors think to use them? Usually ¹⁹F-MRI is used for tracking of therapeutic cells in vivo or for monitoring inflammation progression, maybe this system is not suitable to these applications as I expect a low cellular uptake (too low sensitivity for in vivo imaging). Maybe such a system would be better suited to drug delivery applications. I think that the authors should comment on this somewhere in the text.

Reviewer #2:

Remarks to the Author:

The manuscript "Biodegradable Polyphosphoester Micelles Act as Both Background-free ³¹P Magnetic Resonance Imaging Agents and Drug Nanocarriers" describes a novel approach for in vivo imaging of introduced nanoparticles based on their rich ³¹P-content, which is detected by MRI without contributions of MR signals of endogenous ³¹P-content. Throughout their manuscript, Koshkina, Fogel, Frederik and their colleagues demonstrate the key aspects of their development including the synthetic strategies to obtain an ultimate ³¹P copolymer that self assemble into well-defined micelles and allow favorable MRI characteristic. The ³¹P-MRI properties of the preferred polyphosphoester micelles were studied before demonstrating them in vivo in Manduca sexta. The hydrophobic core of the obtained amphiphilic micelles was used to encapsulate and deliver a PROTAC drug payload demonstrating also the performances of these biodegradable formulation in a therapeutics scenario. Overall, this is a very elegant manuscript showing a rational design and application of a novel biocompatible, degradable polyphosphoesters for background-free heteronuclear (³¹P) MRI studies. Nevertheless, although presenting high-quality research, which is a requirement for publications in Nature Communications, I was not convinced that this manuscript shows the novelty and the important significant advances in the field of molecular MRI, for being accepted to this journal at the stage it stands now.

As a general comment, I would recommend to re-write the introduction, which is not fluently read.

Below please find my specific comments:

1. Although the authors overcame, very elegantly, any potential chemical shift artifacts in in vivo ³¹P-

MRI studies with their newly design formulation, the sensitivity of ³¹P-MR is an intrinsic property that cannot be solved only by a rationale design. The MR sensitivity of ³¹P, even at its 100% natural abundance, is expected to be lower by orders of magnitude when compared to that of ¹⁹F, which is nowadays the most used non-¹H nuclei for background free MRI studies. Therefore, without introducing an MRI signal amplification strategy (i.e., hyperpolarization, significant T1 shortening, etc.) I don't see the practical application of this approach. It was not clear what is the sensitivity of the approach. Or in other words, how many ³¹P spins are needed in an imaging voxel in order to obtain a detectable ³¹P-MRI signal at a given scan time. The authors mentioned, in relate to other ³¹P-based formulations: "To overcome the lower sensitivity of the ³¹P-nucleus, we developed biodegradable nanocarriers that are based on bioinspired polymers with a high ³¹P-content, i.e. polyphosphoesters (PPEs)." The low sensitivity of ³¹P-MR is a fact. I do not see how biodegradable polymers can overcome this challenge. This point must be addressed in order to allow readers to evaluate the potentiality of the approach in other scenarios, beyond the one shown here.

2. In a relation to the previous comment, the authors stated: "In spite of the lower sensitivity of the ³¹P nucleus, compared with ¹⁹F, we selected a dose typical for intravenous injection of ¹⁹F MRI agents (15 mg in 100 μL)." This is not a proper comparison. What was the weight of the formulation per Kg body of the subject. A dose of 15 mg in 100 μL used in *Manduca sexta* caterpillars could be way too high when compare to doses used of imaging agents in rodents. This point should be clarified and a comparison to ¹⁹F with which the authors have a great experience, should be properly shown.

3. The sentence "We realized thus far impossible imaging of ³¹P, overcoming the low gyromagnetic ratio of ³¹P-nucleus by 30 tailoring colloids' structural features" in the abstract is not clear and should be rephrased. In addition, the authors cited previous works that demonstrated ³¹P-MRI so this particular sentence should be toned down. The same stands for "Our solution introduces thus far impossible imaging of ³¹P-nucleus, which we realized by tailoring the structural properties of phosphorus-containing polymers."

4. "the low sensitivity of 7 % compared to proton." Should be replaced with "as compared to that of ¹H-MR."

5. "Yet, most of biomolecules do not give rise to strong NMR signals due to unfavorable properties, such as coupling, multiple signals and short relaxation times." Coupling should be replaced with J-coupling. "Short relaxation times" is a too general statement. While short T2 is indeed unfavorable, short T1 is an advantage.

6. "With these T2-times, both nanostructures were successfully imaged with ³¹P MRI within acquisition time of 17 min (TAcq; Fig. 1C), proving our strategy". This sentence is meaningless without reporting on the experimental parameters used such as the ³¹P concentrations, the in-plane resolution and the slice thickness used to obtain these and other data sets in the manuscript.

7. In Figures 1-3, which show MRI data, the captions should contain the experimental parameters used to obtain the ³¹P-MR images. This includes, magnetic field, ³¹P concentration and relevant acquisition parameters such as in plane resolution and slice thickness and scan time if not shown already.

8. "We assume that the previously reported stealth effect of EtPPn25 prolonged the circulation time; thus, only a small fraction was excreted or degraded." This is a speculation. If the authors could study the fate of their formulation (e.g., by ³¹P-MRS) this should be performed. Otherwise, I recommend to remove this sentence.

9. "However, HeLa cells are generally less responsive to cytotoxic effects of degradation of bromodomain and extra-terminal proteins caused by ARV-825 treatment,³⁰ which explains a higher remaining viability observed here". Given this sentence, it is not clear to me why did the authors used HeLa cells at first place and did not choose a proper cell-line for their demonstration. As the purpose of this part was to demonstrate the performances of their novel formulation as a drug carrier to improve drug performances, a better platform should be used for the examination.

10. Conclusion: "We developed biocompatible, biodegradable micelles...overcoming the low sensitivity of the ³¹P-nucleus." This sentence should be rephrased. The low sensitivity is not of the nucleus but of ³¹P-MRI.

Reviewer #3:

Remarks to the Author:

The authors present a novel type of MRI contrast agent based on polyphosphoester micelles. These feature a detectable ^{31}P MR signal and have the potential to function as nanocarriers, as is demonstrated by encapsulation of the drug PROTAC. The final formulation of the novel contrast agent has favourable relaxation times for MRI, and is shown to provide signal after injection into fruit or *Manduca sexta*. Synthesis of the micelles and characterization of stability and MR-relevant properties is described in sufficient detail. Conservation of the drug action of PROTAC after encapsulation was demonstrated in one experiment on HeLa cells. Metabolic degradation of the micelles by *M. sexta* was shown, however, not characterized in more detail. In summary, the manuscript shows that the novel MR contrast agent is a versatile and promising candidate for use in biomedical imaging and potentially also in the clinics. However, there are several points that need to be addressed:

1. Major weakness of the paper is that the authors do not provide any measure or estimate, how sensitive detection of this novel contrast agent can be. To this end, a detailed assessment of signal/noise ratio per unit time and unit volume for different concentrations and ideally at different magnetic field strengths would be required. These values should be compared and discussed with achievable SNR for other (eg., ^{19}F) contrast agents.
2. Assessment of the temperature dependence of relaxation time and detectable signal (SNR) is required, if biomedical or clinical applications are targeted. At a minimum relaxation times and signal (SNR) must be characterized at 37 °C. What was the temperature for the *M. sexta* measurements?
3. Presented MR images in the figures lack details: spatial resolution and scan time should be stated throughout. Scale bars should be added to all MR images. Color bars (with units) should be added in Fig. 2 and 3.
4. Relaxation times are given without errors or standard deviations. This may be adequate in analytical chemistry. However, for contrast agents for imaging, reproducibility for different productions should be assessed and the remaining errors stated.
5. It is stated that blood was drawn from mice. Specify and add number of animal use protocol.
6. FACS for immune cells is mentioned but not described further.

REVIEWER COMMENTS

Reviewer #1 (Remarks to the Author):

The work from Koshkina et al. is about the development of biodegradable polyphosphonate micelles for ^{31}P -MRI. These polymers form in aqueous solution small micelles and a tailored design of the polymer skeleton with a low glass transition T_g guarantee T_1 and T_2 suitable for imaging. This work is extremely innovative and although preliminary the smart chemistry linked to the advanced acquisition methods allow to overcome the sensitivity problems usually associated with ^{31}P -MRI. This work can open up a new alternative/complementary method for in vivo imaging to ^{19}F -MRI, which is needed as most of the perfluorinated compounds used as MRI probes can raise sustainability issues. This work deserves publication in Nature Communication after minor revisions reported below.

Response:

We appreciate that the reviewer found our manuscript extremely innovative and suitable for publication in Nature Communications.

- it would be important to report at least for the best polymer (gradient) a ^{31}P -MRI concentration study for $c > c_{mc}$ of the micellar suspension to show linearity of the signal and also the minimal concentration for which is possible to obtain a suitable SNR.

Response:

We have now measured ^{31}P MRI of a dilution series, and included the results in Figure S11. As expected the signal intensity linearly correlates with SNR: The lowest detectable concentration was 0.1 wt% PPn_{GRAD} micelles, and reasonable signal was obtained at 0.5 wt%. Note that this concentration was still detectable with mCSSL sequence which is modified RARE for detection of multiple peaks. The detection limits for spectroscopic approaches are expected to be even lower.

The following data were added:

Fig. S11. Determination of ^{31}P MRI SNR. (A) ^{31}P MRI of micelles in water containing 4, 1, 0.5 and 0.1 wt.1% PPn_{GRAD} (counterclockwise from top left). (B) SNR ($n=3$) plotted versus concentration of PPn_{GRAD} with a linear fit ($R^2 = 0.99$). FOV $20 \times 20 \text{ mm}^2$, Matrix 64×64 , Slice thickness 8 mm, phantom diameter 0.97 cm, TAcq 21 min 20 sec.

- it would be important for the reader to know the number of ^{31}P atoms/mL for the reported MRI experiments rather than the w/V concentration of the polymer (also for a better comparison with ^{19}F -MRI). They comment that they used the same "concentration" as for the probe in ^{19}F -MRI, but it is not clear if it is the same P/F concentration or the same polymer concentration, which would be not relevant as the Mw is different (moreover, most ^{19}F -MRI are molecular probes). This should be better explained in the text.

Response:

This is a good point- we have now added the concentration of ^{31}P in mol in addition to wt% in the main manuscript (see updated figure captions).

We have indeed decided to use the same polymer concentration, as it usually would determine the maximal injectable dose, as now also specified in the caption to figure 2.

- Did they perform ^{19}F -MRI measurements in Manduca for comparison? If they have them it would be important to introduce them in the ESI.

Response:

We did not perform these measurements, as our PPN are not directly comparable to PFCE emulsions: the different size and surface properties will influence the distribution, uptake and clearance. We are planning to perform an extended biodistribution study in the future where we systematically address these questions, particularly developing new ^{19}F formulations better comparable with our colloids, but it goes beyond the scope of current work.

- these micelles are very small, for which application the authors think to use them? Usually ^{19}F -MRI is used for tracking of therapeutic cells in vivo or for monitoring inflammation progression, maybe this system is not suitable to these applications as I expect a low cellular uptake (too low sensitivity for in vivo imaging). Maybe such a system would be better suited to drug delivery applications. I think that the authors should comment on this somewhere in the text.

Response:

The polymer platform presented in this study can be tailored to different applications. Here, we target an earlier more fundamental phase of research before focusing on specific application, and provide the first proof for several highly innovative elements: (1) The first MRI of polymers using the signal from their backbone and NOT an actual label. (2) The first proof that MRI of ^{31}P with spin echo sequences becomes possible through our chemistry. (3) First proof that ^{31}P MRI can help to monitor the changes in polymer properties in vivo, e.g. their degradation.

Polyphosphonates belong to the polymer class of polyphosphoesters which are comparably new polymers handled as promising for various biomedical applications, including drug delivery already included here as an example, and tissue regeneration and also cell labeling and activation. This work provides the first key step with the introduction of MRI-traceable polymers and will enable tailoring the polymers to all these applications in the future.

We have now modified several sections in the manuscript to emphasize this point. As we rewrote major part of the introduction, we do not list them here but only in the manuscript.

Reviewer #2 (Remarks to the Author):

The manuscript "Biodegradable Polyphosphoester Micelles Act as Both Background-free ^{31}P Magnetic Resonance Imaging Agents and Drug Nanocarriers" describes a novel approach for in vivo imaging of introduced nanoparticles based on their rich ^{31}P -content, which is detected by MRI without contributions of MR signals of endogenous ^{31}P -content. Throughout their manuscript, Koshkina, Flogel, Frederik and their colleagues demonstrate the key aspects of their development including the synthetic strategies to obtain an ultimate ^{31}P copolymer that self assemble into well-defined micelles and allow favorable MRI characteristic. The ^{31}P -MRI properties of the preferred polyphosphoester micelles were studied before demonstrating them in vivo in Manduca sexta. The hydrophobic core of the obtained amphiphilic micelles was used to encapsulate and deliver a PROTAC drug payload

demonstrating also the performances of these biodegradable formulation in a therapeutics scenario. Overall, this is a very elegant manuscript showing a rational design and application of a novel biocompatible, degradable polyphosphoesters for background-free heteronuclear (^{31}P) MRI studies. Nevertheless, although presenting high-quality research, which is a requirement for publications in Nature Communications, I was not convinced that this manuscript shows the novelty and the important significant advances in the field of molecular MRI, for being accepted to this journal at the stage it stands now. As a general comment, I would recommend to re-write the introduction, which is not fluently read.

Response:

We are grateful that the reviewer finds our manuscript elegant and emphasizes the novelty. Following the reviewer's suggestion, we have modified the abstract and introduction stressing out the impact of this on the development of biomedical polymers- the introduction of biodegradable polymers that are directly traceable with MRI from their backbone.

The changes made in to the introduction are too extensive to list them here, as we rewrote major part of the introduction following the suggestion. They are all highlighted in the manuscript.

Below please find my specific comments:

1. Although the authors overcame, very elegantly, any potential chemical shift artifacts in in vivo ^{31}P -MRI studies with their newly design formulation, the sensitivity of ^{31}P -MR is an intrinsic property that cannot be solved only by a rationale design. The MR sensitivity of ^{31}P , even at its 100% natural abundance, is expected to be lower by orders of magnitude when compared to that of ^{19}F , which is nowadays the most used non- ^1H nuclei for background free MRI studies. Therefore, without introducing an MRI signal amplification strategy (i.e., hyperpolarization, significant T1 shortening, etc.) I don't see the practical application of this approach. It was not clear what is the sensitivity of the approach. Or in other words, how many ^{31}P spins are needed in an imaging voxel in order to obtain a detectable ^{31}P -MRI signal at a given scan time. The authors mentioned, in relate to other ^{31}P -based formulations: "To overcome the lower sensitivity of the ^{31}P -nucleus, we developed biodegradable nanocarriers that are based on bioinspired polymers with a high ^{31}P -content, i.e. polyphosphoesters (PPEs)." The low sensitivity of ^{31}P -MR is a fact. I do not see how biodegradable polymers can overcome this challenge. This point must be addressed in order to allow readers to evaluate the potentiality of the approach in other scenarios, beyond the one shown here.

Response:

The suggestions to implement T1 shortening strategy and combine it with signal amplification approach are indeed interesting to address in the future but are not required for this pioneering publication that focuses on the earlier stages of the development of new materials.

This study provides the first proof of novel polymers that (1) can be imaged ^{31}P MRI, opening the route for the development of polymer materials for a broad range of applications (2) are biodegradable polymers and (3) to our knowledge also the first time that a spin-echo sequence and not much more sensitive but time-consuming spectroscopic approaches are used to detect ^{31}P . We have now rewritten the introduction putting the focus on the above points (the changes are too extensive to be listed here but are all visible in highlighted manuscript).

Coming to the comparison of the sensitivity of ^{31}P and to ^{19}F , the sensitivity is not the sole factor that determines the eventual SNR. In vivo, the biodistribution and the concentration in the target tissues, together with relaxation times govern the eventual signal intensity, and not

the sensitivity alone. While the sensitivity of ^{19}F nucleus is higher, the ^{19}F emulsions often show increased unwanted accumulation in the liver, which is expected due to the poor miscibility of PFC. Moreover, PFC are non-degradable and extremely stable, thus they can accumulate in organs for months which is a big disadvantage for biomedical use. At the same time, PFC emulsions suffer from various colloidal stability issues that arise from the amphiphobic character of PFC which makes immiscible with water and with organic solvents. In short: higher sensitivity of pure compound does not mean higher imaging intensity of the target organ.

Clearly, longer-term, an *in vivo* study in an actual disease model comparing targeted and non-targeted agents would be necessary to systematically compare the performance but goes beyond the scope of current already very extensive work which is at the earlier stage of material development.

To further address the sensitivity, we have measured MRI of micelles at different concentrations and included the results in the SI. The lowest detectable concentration was 0.1 wt.-% of polymer, and sufficient SNR was obtained at which is in a comparable range to some fluorinated polymers used for example for tumor imaging (cf. e.g. Zhang et al. ACS Nano 2018, 12, 9162).

Fig. S11. Determination of ^{31}P MRI SNR. (A) ^{31}P MRI of micelles in water containing 4, 1, 0.5 and 0.1 wt.1% PPn_{GRAD} (counterclockwise from top left). (B) SNR (n=3) plotted versus concentration of PPn_{GRAD} with a linear fit ($R^2 = 0.99$). FOV 20x20 mm², Matrix 64x64, Slice thickness 8 mm, TAcq 21 min 20 sec.

The following sentence was further added in the main manuscript (p.8):

“The signal intensity decreased linearly, when the PPn_{GRAD} dispersion was diluted (Fig. S11). The lowest detectable concentration of polymer was comparable to the one reported for fluorinated polymers previously,¹³ indicating sufficient signal intensity for in vivo studies.”

2. In a relation to the previous comment, the authors stated: “In spite of the lower sensitivity of the ^{31}P nucleus, compared with ^{19}F , we selected a dose typical for intravenous injection of ^{19}F MRI agents (15 mg in 100 μL).” This is not a proper comparison. What was the weight of the formulation per Kg body of the subject. A dose of 15 mg in 100 μL used in *Manduca sexta* caterpillars could be way too high when compare to doses used of imaging agents in rodents. This point should be clarified and a comparison to ^{19}F with which the authors have a great experience, should be properly shown.

Response:

Manduca sexta display a comparable weight and same hemolymph volume as mice, as we already explained in the main manuscript. The article that introduced *M. sexta* as alternative

animal model has been published after the submission of this manuscript, and is now cited in the main manuscript further supporting our statement (cf. A. G. Windfelder et al., Nat. Commun 2022, 13, 7216, and iScience 2023, 196801). Thus, injecting *M. Sexta* with the same concentration as used for fluorinated agents provides a perfect comparison for this proof of concept study, and reduces the animal testing in accordance with the legislation of the European Union. The concentration by weight was selected as in case of colloidal agents, such as emulsions and polymer micelles, it determines the possible injectable dose. As already stated in the previous response, several parameters can affect the biodistribution and the resulting signal intensity which we plan to address in the future.

3. The sentence “We realized thus far impossible imaging of ^{31}P , overcoming the low gyromagnetic ratio of ^{31}P -nucleus by 30 tailoring colloids’ structural features” in the abstract is not clear and should be rephrased. In addition, the authors cited previous works that demonstrated ^{31}P -MRI so this particular sentence should be toned down. The same stands for “Our solution introduces thus far impossible imaging of ^{31}P -nucleus, which we realized by tailoring the structural properties of phosphorus-containing polymers.”

Response:

We have edited the abstract and introduction with an emphasis on the development of ^{31}P MRI-traceable polymer materials which opens a plethora of opportunities for different biomedical applications.

We would like to bring to reviewer’s attention that to our knowledge this publication is the first report of ^{31}P MRI using a spin echo sequence and not spectroscopic approaches (MRSI) which could not be realized so far due to the poor imaging characteristics of ^{31}P containing biomolecules.

The abstract now starts with:

“We developed a polymer platform uniquely traceable with heteronuclear magnetic resonance imaging (MRI) via ^{31}P in polymer backbone. Monitoring polymers in vivo is essential for applications such as delivery of therapeutics and tissue regeneration.”

The sentences:

„As a solution we introduce polyphosphoester colloids for heteronuclear MRI using ^{31}P -nucleus. We realized thus far impossible imaging of ^{31}P , overcoming the low gyromagnetic ratio of ^{31}P -nucleus by tailoring colloids’ structural features.“

Were changed to:

“Here, we introduce biodegradable, biocompatible polyphosphoester colloids for ^{31}P MRI, solving fundamental issues in MRI of ^{31}P , including intrinsic background and low sensitivity.”
In the last sentence of the abstract “, opening a route to MRI-traceable polymers” was added to put the emphasis on MRI-traceable polymer materials for which our polymers provide a unique approach.

Similar changes were made in the introduction of the manuscript, all visible in the highlighted version; we do not list them here to keep it short.

4. “the low sensitivity of 7 % compared to proton.” Should be replaced with “as compared to that of ^1H -MR.”

Response:

Agreed- we have changed

“however, the development of agents for ³¹P MRI with rapid acquisition sequences has been impossible thus far due to the low sensitivity of 7 % compared to proton.”

To:

“However, the development of background-free ³¹P MRI agents and ³¹P MRI-traceable materials has been hampered by several factors, including the intrinsic background from natural phosphates, the low gyromagnetic ratio that results in a low MR sensitivity of ³¹P of 7 % compared to ¹H, and other unfavorable MR characteristics”

5. “Yet, most of biomolecules do not give rise to strong NMR signals due to unfavorable properties, such as coupling, multiple signals and short relaxation times.” Coupling should be replaced with J-coupling. “Short relaxation times” is a too general statement. While short T2 is indeed unfavorable, short T1 is an advantage.

Response: *We agree and have changed it to “J-coupling” and “too short transverse and too long longitudinal relaxation times”*

6. “With these T2-times, both nanostructures were successfully imaged with ³¹P MRI within acquisition time of 17 min (TAcq; Fig. 1C), proving our strategy”. This sentence is meaningless without reporting on the experimental parameters used such as the ³¹P concentrations, the in-plane resolution and the slice thickness used to obtain these and other data sets in the manuscript.

Response:

We agree and have modified the sentence and added that polymers were imaged using a spin-echo sequence.

The sentence:

“With these T₂-times, both nanostructures were successfully imaged with ³¹P MRI within acquisition time of 17 min (TAcq; Fig. 1C).”

Was changed to:

“As a result of improved T₂-times, both nanostructures were successfully imaged with ³¹P MRI using a rapid spin-echo sequence within acquisition time of 17 min (TAcq; Fig. 1C, see methods for further details).”

The experimental details including FOV and concentration were further added in the figure caption.

7. In Figures 1-3, which show MRI data, the captions should contain the experimental parameters used to obtain the ³¹P-MR images. This includes, magnetic field, ³¹P concentration and relevant acquisition parameters such as in plane resolution and slice thickness and scan time if not shown already.

Response:

We thank the reviewer for this remark and have adjusted the figure captions and added the magnetic field, FOV and concentration.

8. “We assume that the previously reported stealth effect of EtPPn25 prolonged the circulation time; thus, only a small fraction was excreted or degraded.” This is a speculation. If the authors could study the fate of their formulation (e.g., by ³¹P-MRS) this should be performed. Otherwise, I recommend to remove this sentence.

Response:

The figure demonstrates that micelles remain in the circulation by comparing MRI after injection and after 24 h, and it is known from our previous work (Nat. Nanotechnol. 2016, Angew. Chem. 2018) that hydrophilic polyphosphoesters, including Et-PPn display the stealth effect. Therefore, the statement about stealth effect builds on our results, literature and is appropriate, as it starts with “we assume”.

We agree with the reviewer that it would be interesting and necessary for disease imaging to study the biodistribution in the future. We have therefore modified this section:

“After 24 h, the ³¹P-signal was still located in the hemolymph and decreased only slightly (Fig. 3C). We assume that the previously reported stealth effect of Et-PPn³² prolonged the circulation time; thus, only a small fraction was excreted or degraded.”

Was changed to:

After 24 h, the ³¹P-signal was still located in the hemolymph and decreased only slightly (Fig. 3C). Thus, only a small fraction was excreted or degraded; the biodistribution should be studied in the future in different animal models. We assume that the previously reported stealth effect of Et-PPn³² prolonged the circulation time, allowing one to detect the micelles in hemolymph after 24 h.

9. “However, HeLa cells are generally less responsive to cytotoxic effects of degradation of bromodomain and extra-terminal proteins caused by ARV-825 treatment,³⁰ which explains a higher remaining viability observed here“. Given this sentence, it is not clear to me why did the authors used HeLa cells at first place and did not choose a proper cell-line for their demonstration. As the purpose of this part was to demonstrate the performances of their novel formulation as a drug carrier to improve drug performances, a better platform should be used for the examination.

Response:

Important point- initially we used HeLa because they are commonly used in studies with cancer nanomedicines. Our results show that the polymers stabilize a hydrophobic, PROTAC which is insoluble in water and demonstrate a clear increase decrease. Therefore, we agree with the reviewer that this sentence might be misleading and removed this part of discussion.

10. Conclusion: “We developed biocompatible, biodegradable micelles...overcoming the low sensitivity of the ³¹P-nucleus.” This sentence should be rephrased. The low sensitivity is not of the nucleus but of ³¹P-MRI.

Response: We agree and have changed the conclusion section accordingly.

Reviewer #3 (Remarks to the Author):

The authors present a novel type of MRI contrast agent based on polyphosphoester micelles. These feature a detectable ³¹P MR signal and have the potential to function as nanocarriers, as is demonstrated by encapsulation of the drug PROTAC. The final formulation of the novel contrast agent has favourable relaxation times for MRI, and is shown to provide signal after injection into fruit or *Manduca sexta*. Synthesis of the micelles and characterization of stability and MR-relevant properties is described in sufficient detail. Conservation of the drug action of PROTAC after encapsulation was demonstrated in one experiment on HeLa cells. Metabolic degradation of the micelles by *M. sexta* was shown, however, not characterized in more detail. In summary, the manuscript shows that the novel MR contrast agent is a versatile and promising candidate for use in biomedical imaging and potentially also in the clinics. However, there are several points that need to be addressed:

We are pleased that the reviewer finds our MRI micelles versatile and highly promising

1. Major weakness of the paper is that the authors do not provide any measure or estimate, how sensitive detection of this novel contrast agent can be. To this end, a detailed assessment of signal/noise ratio per unit time and unit volume for different concentrations and ideally at different magnetic field strengths would be required. These values should be compared and discussed with achievable SNR for other (eg., 19F) contrast agents.

Response:

We agree and added novel measurements using different concentrations of the new formulations in the SI (Fig. S11) showing that 0.1 wt.-% polymer still can be detected with mCSSI sequence and 0.5 wt.-% provide a satisfactory SNR.

2. Assessment of the temperature dependence of relaxation time and detectable signal (SNR) is required, if biomedical or clinical applications are targeted. At a minimum relaxation times and signal (SNR) must be characterized at 37 °C. What was the temperature for the *M. Sexta* measurements?

Response:

The measurements in *M. Sexta* were done at room temperature; *M. Sexta* are poikilotherm animals. Nevertheless, we fully agree with the reviewer that the measurements at 37 °C are important and added them in Table S3.

Table S3. Comparison of NMR relaxation times of different polymer batches (158 MHz, cf. Error! Reference source not found. for other polymer properties).

Batch	T ₁ Et-PPn (s)	T ₂ Et-PPn (s)	T ₁ Ph-PPn (s)	T ₂ Ph-PPn (s)
Ph-grad-Et-PPn batch 1 (PPn _{GRAD}) 25 °C	1.8	0.97	0.87	0.11
Ph-grad-Et-PPn batch 2 (PPn _{GRAD}) 25 °C	1.7	0.83	0.88	0.13
Ph-grad-Et-PPn batch 2 (PPn _{GRAD}) at 37 °C	2.1	0.74	0.84	0.22
Ph-grad-Et-PPn batch 3 (PPn _{GRAD}) 25 °C	1.8	0.98	0.87	0.14
Average (25 °C) ± standard deviation	1.8 ± 0.06	0.93 ± 0.07	0.87±0.05	0.13 ±0.01

3. Presented MR images in the figures lack details: spatial resolution and scan time should be stated throughout. Scale bars should be added to all MR images. Color bars (with units) should be added in Fig. 2 and 3.

Response:

Great observation- the scan time and all other parameters were added where they were still missing, along with the scale bar, FOV and color bar on Fig. 2.

4. Relaxation times are given without errors or standard deviations. This may be adequate in analytical chemistry. However, for contrast agents for imaging, reproducibility for different productions should be assessed and the remaining errors stated.

Response:

It is indeed common practice in chemical disciplines, and for the majority of the commercial chemicals to provide analysis for one specific batch to address potential batch to-batch variations, e.g. certificate of analysis for commercial products is always provided for a

specific batch. Following this practice, we already included the results of different polymerizations in the initial submission and listed which data was used on which figure (cf. Table S1)

To address the reviewers concern, we have now added the results relaxation times of several batches in the SI, Table S3, along with average and standard deviation.

Table S3. Comparison of NMR relaxation times of different polymer batches (158 MHz, cf. Error! Reference source not found. for other polymer properties).

Batch	T ₁ Et-PPn (s)	T ₂ Et-PPn (s)	T ₁ Ph-PPn (s)	T ₂ Ph-PPn (s)
Ph-grad-Et-PPn batch 1 (PPn _{GRAD}) 25 °C	1.8	0.97	0.87	0.11
Ph-grad-Et-PPn batch 2 (PPn _{GRAD}) 25 °C	1.7	0.83	0.88	0.13
Ph-grad-Et-PPn batch 2 (PPn _{GRAD}) at 37 °C	2.1	0.74	0.84	0.22
Ph-grad-Et-PPn batch 3 (PPn _{GRAD}) 25 °C	1.8	0.98	0.87	0.14
Average (25 °C) ± standard deviation	1.8 ± 0.06	0.93 ± 0.07	0.87±0.05	0.13 ±0.01

5. It is stated that blood was drawn from mice. Specify and add number of animal use protocol.

Response: The protocol number has been added. Note that according to German animal protection law no animal license is required to withdraw blood from the animal, when animals are under license for a different experiment which was the case here.

6. FACS for immune cells is mentioned but not described further.

The technical details on FACS were added in the materials and methods

Reviewers' Comments:

Reviewer #1:

Remarks to the Author:

The revised version deserves publication in Nature Comm.

Reviewer #2:

Remarks to the Author:

The authors addressed most comments of the three reviewers, and I congratulate them for their great scientific work.

However, I still think comparing the sensitivity of ^{31}P formulations to that of ^{19}F formulations should be experimentally demonstrated. This point was raised independently by each of the three reviewers. Given that the research team (especially Prof. Flogel, who is one of the pioneers and leaders in that field) has all the needed capabilities and expertise to perform world-class ^{19}F -MRI experiments, this point left me disappointed, and I encourage the authors to perform such a simple (even of phantoms) experiment. Such a comparison will help readers evaluate the potentiality of the approach to be further implemented for other applications.

Reviewer #3:

Remarks to the Author:

The authors have sufficiently addressed all points I had raised. All missing data have been added. The manuscript has improved substantially. I have no further concerns that abrogate publication.

Cornelius Faber, University of Münster, Germany

Point by point response

We kindly thank all reviewers and the editor- the remaining minor comment has been fully addressed

Reviewer #2 (Remarks to the Author):

The authors addressed most comments of the three reviewers, and I congratulate them for their great scientific work.

However, I still think comparing the sensitivity of ^{31}P formulations to that of ^{19}F formulations should be experimentally demonstrated. This point was raised independently by each of the three reviewers. Given that the research team (especially Prof. Fogel, who is one of the pioneers and leaders in that field) has all the needed capabilities and expertise to perform world-class ^{19}F -MRI experiments, this point left me disappointed, and I encourage the authors to perform such a simple (even of phantoms) experiment. Such a comparison will help readers evaluate the potentiality of the approach to be further implemented for other applications.

Response

Data has been included in Supplementary Fig. 11